# ADBM: Adversarial Diffusion Bridge Model for Denoising of 3D Point Cloud Data

**DOI:** 10.3390/s25175261

**Published:** 2025-08-24

**Authors:** Changwoo Nam, Sang Jun Lee

**Affiliations:** Division of Electronic Engineering, Jeonbuk National University, 567 Baekje-daero, Deokjin-gu, Jeonju 54896, Republic of Korea; cw.nam@jbnu.ac.kr

**Keywords:** deep learning, diffusion model, adversarial training, generative model, 3D point cloud denoising

## Abstract

We address the task of point cloud denoising by leveraging a diffusion-based generative framework augmented with adversarial training. While recent diffusion models have demonstrated strong capabilities in learning complex data distributions, their effectiveness in recovering fine geometric details remains limited, especially under severe noise conditions. To mitigate this, we propose the Adversarial Diffusion Bridge Model (ADBM), a novel approach for denoising 3D point cloud data by integrating a diffusion bridge model with adversarial learning. ADBM incorporates a lightweight discriminator that guides the denoising process through adversarial supervision, encouraging sharper and more faithful reconstructions. The denoiser is trained using a denoising diffusion objective based on a Schrödinger Bridge, while the discriminator distinguishes between real, clean point clouds and generated outputs, promoting perceptual realism. Experiments are conducted on the PU-Net and PC-Net datasets, with performance evaluation employing the Chamfer distance and Point-to-Mesh metrics. The qualitative and quantitative results both highlight the effectiveness of adversarial supervision in enhancing local detail reconstruction, making our approach a promising direction for robust point cloud restoration.

## 1. Introduction

Point cloud denoising is critical for enhancing data quality in applications where accurate spatial representation directly impacts system performance and user accessibility. Point clouds acquired via LiDAR, depth sensors, or photogrammetry frequently contain noise from environmental interference, sensor limitations, or motion artifacts. This degradation is especially critical in accessibility applications, such as assistive navigation, where noisy inputs cause errors in object detection [1,2,3] and scene reconstruction [4]. Also, the presence of noise can obscure fine geometric details and lead to inaccurate shape representations, which are especially problematic for applications requiring high-precision measurements. As the reliance on 3D point cloud data also continues to grow across diverse fields such as robotics [5,6], urban mapping [7,8], and medical imaging [9,10], the demand for robust and effective denoising techniques is becoming increasingly important.

Traditional 3D point cloud denoising approaches [11,12,13,14] have mainly relied on geometric priors and statistical optimization. These approaches demonstrated measurable denoising efficacy under controlled conditions, particularly for Gaussian-type noise distributions. However, they consistently struggled with structural oversimplification in real-world scenarios, where rigid smoothing operators erode fine features like edges and corners, degrading geometric fidelity. Also, non-Gaussian noise from LiDAR or other sensors caused performance collapse, while iterative optimization hindered real-world deployment. These limitations have prompted a shift toward learning-based denoising approaches to adaptively model complex noise patterns while maintaining geometric fidelity.

Recent years have seen generative models, particularly diffusion models, emerge as powerful tools for 3D point cloud data synthesis and restoration [15,16,17]. By iteratively refining their understanding of complex data distributions, these models achieve high-fidelity reconstruction of noisy inputs through the structured denoising process. However, traditional diffusion approaches suffer from slow sampling speeds, sampling trajectory design inefficiencies, and instability when handling complex noise distributions. Diffusion bridges [18,19,20,21] address these gaps by predicting a direct probabilistic pathway between noisy and clean data distributions, through mitigating the constraints on the prior distribution. While the direct pathway offers improved sampling efficiency and stability, achieving optimal denoising performance, particularly against complex and unknown noise patterns, necessitates a more adaptive and self-improving mechanism.

Inspired by the success of adversarial learning in generative models [22,23,24,25], we propose the Adversarial Diffusion Bridge Model (ADBM), which integrates adversarial supervision into the diffusion bridge framework to enhance 3D point cloud denoising. Specifically, a lightweight discriminator is incorporated into the training pipeline to compel the diffusion bridge model to generate outputs that are not only distributionally close to clean data but also perceptually realistic. As shown in Figure 1, ADBM effectively restores clean shapes from severely noisy inputs across various object categories. The adversarial signal complements the original diffusion bridge objective, providing an additional learning signal that facilitates the recovery of fine geometric details, particularly under complex or non-Gaussian noise conditions. We validate ADBM on PC-Net [15] and PU-Net [26], 3D object-level point cloud datasets. The experimental results demonstrate that ADBM consistently outperforms existing state-of-the-art denoising methods in terms of both fidelity and generalization. In summary, the main contributions of this paper are as follows:We propose ADBM, a novel denoising framework that integrates adversarial learning into a diffusion bridge model, enhancing robustness and generation quality for 3D point cloud restoration.We design an adversarial training objective specifically formulated for diffusion-based point cloud denoising, which reconstructs fine-grained geometric details of the 3D point cloud.We perform comparative evaluations on the PU-Net and PC-Net datasets, using the latter solely for testing, and demonstrate that ADBM achieves state-of-the-art denoising performance with strong generalization across unseen objects categories and varying resolutions.

The remainder of this paper is organized as follows: Section 2 reviews relevant literature on point cloud denoising. Section 3 introduces the proposed method. Section 4 and Section 5 present the experimental results and conclusions, respectively.

## 2. Related Work

### 2.1. Traditional Denoising Methods

Traditional methods for 3D point cloud denoising mainly leverage geometric priors and local statistics to suppress noise while preserving structural features. Han et al. [11] proposed a position-guided linear filter for 3D point cloud denoising that significantly improves computational efficiency while preserving geometric features. To preserve sharp features in noisy point clouds, Zheng et al. [12] proposed a guided filter extension that assigns multiple normals to feature points via k-medial skeleton extraction and k-means clustering. To enhance the quality of noisy point sets, Yadav et al. [13] introduced a constraint-based denoising method utilizing a vertex-based normal voting tensor and binary eigenvalue optimization. Their approach iteratively filters vertex normals and updates positions with feature-aware constraints, enabling effective noise removal while preserving geometric sharpness. To address the trade-off between noise removal and feature preservation, Liu at al. [14] developed a two-stage point cloud denoising method that decouples normal filtering from position updating. Their optimization-based framework maintains the underlying geometric structures, achieving high-quality denoising without oversmoothing sharp edges.

### 2.2. Deep Learning-Based Methods

To overcome the limitations of traditional denoising approaches, recent research has shifted toward learning-based methods that leverage neural networks to model complex noise patterns in point clouds. PointCleanNet [15] introduced supervised frameworks that learn mappings from noisy to clean point clouds using regression-based losses. They employ an architecture that explicitly encodes spatial features while incorporating a two-step denoising mechanism to refine predictions iteratively. Another notable approach is score-based point cloud denoising [16], which introduces a probabilistic generative framework based on score matching and Langevin dynamics. By learning a score function that estimates the gradient of the data distribution, this method can denoise corrupted point clouds through iterative updates. However, the stochastic nature and high iteration cost of score-based sampling remain key challenges. More recently, the P2P-Bridge [17] framework proposes a diffusion bridge-based model that constructs a direct probabilistic path between noisy and clean point clouds via a Schrödinger Bridge formulation [19]. This method utilizes a learnable forward diffusion and reverse denoising to generate geometrically consistent reconstructions, offering improved sample efficiency and generation quality.

While P2P-Bridge demonstrates strong performance, it remains limited in adaptively learning discriminative features for real-world noise, due to the absence of an explicit adversarial signal. In this work, we integrate adversarial learning on the diffusion bridge model based on P2P-Bridge to further enhance robustness against diverse noise types.

### 2.3. Adversarial Training Approaches

Recent studies have explored adversarial training to improve the quality and realism of diffusion-based generative models. Ko et al. [22] introduces dual discriminators in the time and frequency domains to enhance speech fidelity in multi-speaker TTS tasks. Zeng et al. [23] leverages semantic priors and adversarial loss for self-supervised shadow removal, enabling structure-preserving generation without paired labels. Liu et al. [24] combines adversarial learning approach with torsion angle priors to ensure biologically valid backbones in protein structure generation. A structure-guided discriminator [25] has also been proposed to fine-tune diffusion models under layout constraints, improving both semantic consistency and image quality. These approaches demonstrate the effectiveness of adversarial signals in guiding diffusion models toward more realistic and task-aligned outputs.

## 3. Methods

We propose ADBM, an adversarial diffusion bridge model based on P2P-Bridge [17], which formulates point cloud denoising as a Schrödinger Bridge problem between clean and noisy distributions. This approach enables efficient sampling of intermediate states without numerically solving stochastic differential equations, by leveraging a Gaussian approximation under a paired data boundary condition. By predicting the underlying noise component, the model iteratively refines the input through a learned reverse process. To improve the perceptual quality of the denoised outputs, we further incorporate an adversarial training objective. A lightweight discriminator is trained to distinguish real clean point clouds from generated samples, providing an additional supervisory signal to guide the denoising network. Figure 2 presents the overall framework.

### 3.1. Diffusion Bridge Training

We formulate point cloud denoising as a Schrödinger Bridge problem, which seeks a stochastic process that interpolates between two marginal distributions: the clean data distribution $p_{\mathrm{data}}\left(x_{0}\right)$ and the noisy prior distribution $p_{\mathrm{prior}}\left(x_{T}\right)$. The goal is to find a path measure $p^{*}\left(x_{0:T}\right)$ that minimizes the Kullback–Leibler divergence from a reference process $p_{\mathrm{ref}}\left(x_{0:T}\right)$ while satisfying the boundary conditions:(1)$$p^{*}\left(x_{0}\right)=p_{\mathrm{data}}\left(x_{0}\right),\;p^{*}\left(x_{T}\right)=p_{\mathrm{prior}}\left(x_{T}\right).$$

Following the formulation proposed in P2P-Bridge, the optimal diffusion path is modeled by a pair of forward and backward stochastic differential equations (SDEs), given, respectively, by(2)$$\begin{array}{cc} dx_{t} & =\left[f\left(x_{t},t\right)+g^{2}\left(t\right)\nabla \log\Psi_{t}\left(x_{t}\right)\right]dt+g\left(t\right)dw_{t},\\ dx_{t} & =\left[f\left(x_{t},t\right)-g^{2}\left(t\right)\nabla \log{\hat{\Psi }}_{t}\left(x_{t}\right)\right]dt+g\left(t\right)d{\bar{w}}_{t}, \end{array}$$
where $f(x_{t},t)$ is a vector-valued drift function, g(t) is a scalar-valued diffusion coefficient controlling the noise, and $w_{t}$, ${\bar{w}}_{t}$ are independent standard Wiener processes. $\Psi_{t}$ and ${\hat{\Psi }}_{t}$ are potential functions associated with the forward and backward processes and these two processes are coupled as follows:(3)$$\Psi_{0}\;{\hat{\Psi }}_{0}=p_{\mathrm{data}},\;\Psi_{T}\;{\hat{\Psi }}_{T}=p_{\mathrm{prior}},\;p_{t}=\Psi_{t}\;{\hat{\Psi }}_{t}.$$

This structure ensures that the marginal density $p_{t}$ interpolates the clean data distribution at t=0 and the noisy prior at t=T, forming a time-consistent probabilistic bridge between the two distributions.

However, directly solving the system of Equation (Equation 2) is not practicable for high-dimensional data. To address this, recent works approximate this bridge under a paired data assumption $p\left(x_{0},x_{T}\right)=p_{\mathrm{data}}\left(x_{0}\right)\;p_{\mathrm{prior}}\left(x_{T}|x_{0}\right)$, and assume linear drift with zero external force, i.e., f=0, yielding a tractable Gaussian posterior. Under the assumption of a linear drift f=0 and a known diffusion schedule g(t), the posterior of the latent process $x_{t}$ conditioned on the endpoints $x_{0}$ and $x_{T}$ can be written in closed form as a Gaussian distribution:(4)$$q\left(x_{t}|x_{0},x_{T}\right)=N\left(\mu_{t},\Sigma_{t}\right),$$
where the mean $\mu_{t}$ and the covariance $\Sigma_{t}$ are given by(5)$$\mu_{t}=\frac{{\bar{\sigma }}_{t}^{2}}{{\bar{\sigma }}_{t}^{2}+\sigma_{t}^{2}}x_{0}+\frac{\sigma_{t}^{2}}{{\bar{\sigma }}_{t}^{2}+\sigma_{t}^{2}}x_{T},\;\Sigma_{t}=\frac{\sigma_{t}^{2}{\bar{\sigma }}_{t}^{2}}{{\bar{\sigma }}_{t}^{2}+\sigma_{t}^{2}}\;I,$$
where $\sigma_{t}^{2}=\int_{0}^{t}g^{2}\left(\tau \right)\;d\tau$ and ${\bar{\sigma }}_{t}^{2}=\int_{t}^{1}g^{2}\left(\tau \right)\;d\tau$ represent the accumulated forward and backward variances up to time *t*, respectively. This analytic form enables efficient sampling of intermediate states $x_{t}$ without requiring numerical integration of the SDE. During training, we sample $x_{t}∼q\left(x_{t}|x_{0},x_{T}\right)$, and define the target noise as the residual between the noisy sample and the clean sample as follows:(6)$$\epsilon =\frac{x_{t}-x_{0}}{\sigma_{t}}.$$

The denoiser network $\epsilon_{\theta }\left(x_{t},t\right)$ is trained to predict this noise using MSE loss:(7)$$L_{\mathrm{MSE}}={∥\epsilon_{\theta }\left(x_{t},t\right)-\epsilon ∥}^{2}.$$

This training objective is conceptually aligned with denoising diffusion probabilistic models, but is distinct in that the noise is conditioned on paired clean and noisy samples, following the diffusion bridge model.

### 3.2. Adversarial Training Method

While the diffusion bridge framework optimizes a noise prediction loss based on the Schrödinger Bridge formulation, we further enhance the denoising performance by incorporating an adversarial learning objective. Inspired by GAN-based training schemes [27], we introduce a discriminator network that encourages the generation of samples which are indistinguishable from clean point clouds. Specifically, let $x_{\mathrm{pred}}$ denote the model-generated clean sample obtained via reverse diffusion, and let $x_{\mathrm{gt}}$ denote the corresponding ground-truth clean point cloud. We define a discriminator D(·) that learns to assign high scores to real samples and low scores to generated samples. During each training step, we first sample $x_{t}∼q\left(x_{t}|x_{0},x_{T}\right)$ and use the denoising network $\epsilon_{\theta }$ to estimate $x_{0}^{\mathrm{pred}}$. We then obtain $x_{\mathrm{pred}}$ via reverse sampling. The discriminator is trained to distinguish real clean point clouds from those synthesized by the denoising model. Following the typical GAN formulation, the discriminator loss is defined as(8)$$L_{D}=-E_{x_{\mathrm{gt}}∼p_{\mathrm{data}}}\left[\log D(x_{\mathrm{gt}})\right]-E_{x_{\mathrm{pred}}∼p_{\theta }}\left[\log\left(1-D(x_{\mathrm{pred}})\right)\right].$$

The generator (i.e., the diffusion bridge model) is trained not only to minimize the original noise prediction loss $L_{\mathrm{MSE}}$, but also to fool the discriminator by maximizing its predicted score. This adversarial objective for the generator is defined as(9)$$L_{\mathrm{adv}}=-E_{x_{\mathrm{pred}}∼p_{\theta }}\left[\log D(x_{\mathrm{pred}})\right],$$
which encourages the generator to maximize the discriminator’s belief that $x_{\mathrm{pred}}$ is a real sample. The adversarial signal thus acts as an additional supervisory signal, particularly effective in recovering complex geometric features that are difficult to optimize solely through point-wise regression. To balance the reconstruction and adversarial objectives, we define the final generator loss as a weighted sum:(10)$$L_{G}=L_{\mathrm{MSE}}+\lambda_{\mathrm{adv}}L_{\mathrm{adv}},$$
where $\lambda_{\mathrm{adv}}$ controls the influence of the adversarial signal. This adversarial extension encourages the generator to produce denoised point clouds that not only minimize numerical reconstruction error but also align with the distribution of real clean point clouds.

The procedure of adversarial diffusion bridge training, including noise prediction, adversarial loss computation, and alternating updates of the generator and discriminator is summarized in Algorithm 1. In the training procedure, we employ an $\lambda_{\mathrm{adv}}$ of 0.7 to balance the MSE loss and adversarial objectives.

### 3.3. Implementation

In this work, we adopt the point cloud denoiser network proposed in P2P-Bridge [17] as our backbone denoiser architecture. The model is designed to predict the drift vector field between clean and noisy point clouds, following the Schrödinger Bridge formulation. The denoiser network follows the encoder–decoder structure of PointNet++ [28], consisting of multi-scale set abstraction modules and feature propagation modules.

To facilitate adversarial learning, we introduce a lightweight discriminator network, which is designed to distinguish between ground-truth clean point clouds and denoised samples generated by the diffusion bridge model. It is important to note that the discriminator is only involved during the training phase to provide adversarial feedback to the generator. During inference, the discriminator is removed entirely, and thus the inference time and latency of ADBM are identical to those of the baseline model. The architecture of the discriminator first applies a point-wise encoder composed of two linear layers with ReLU activation and layer normalization, transforming each point into a latent feature. The resulting latent features are then aggregated via average pooling across the point dimension, yielding a global feature vector for each sample. This global representation is further processed by a two-layer MLP to produce a scalar output indicating the realism of the input. Overall, the discriminator contains only 0.07 million parameters, indicating that it is lightweight and adds minimal overhead to the model.
**Algorithm 1:** Training of Adversarial Diffusion Bridge Model
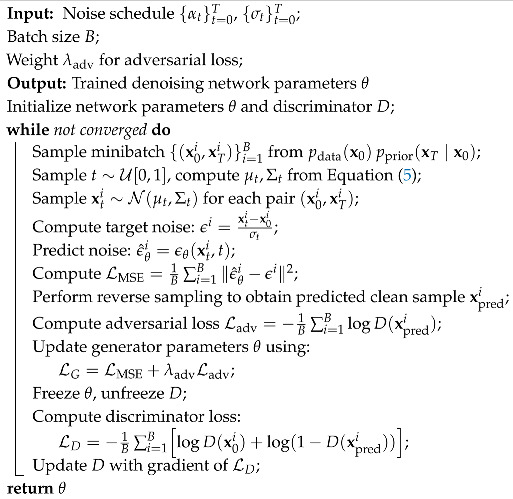



## 4. Experiments

### 4.1. Datasets

We evaluate our method on two benchmark datasets: PU-Net [26] and PC-Net [15]. The PU-Net dataset contains 40 object categories for training and 20 categories for testing. For each object, ground-truth point clouds are provided at three resolutions: 10,000, 30,000, and 50,000 points. To standardize the training input size, we apply farthest-point sampling [28] to extract 2048 points from each noisy input, regardless of its original resolution. This allows the model to be trained on a fixed-size representation while leveraging geometric information from diverse scales. The PC-Net dataset is used solely for testing to assess the generalization ability of the model. It consists of 10 object categories, each provided at three resolutions, totaling 30 test samples. During evaluation, the model outputs a 2048-point cloud, which is then compared to the ground truth using alignment techniques and point-wise distance metrics. This setup allows us to evaluate the denoising performance of the model on both seen and unseen object distributions across varying resolutions.

### 4.2. Evaluation Measure

To quantitatively assess the quality of the denoised point clouds, we adopt two widely used metrics: the Chamfer distance (CD) and Point-to-Mesh (P2M) distance. The CD evaluates the average bidirectional proximity between predicted and ground-truth point sets. It penalizes both missing and redundant points, promoting accurate reconstruction and uniform coverage. Formally, it is defined as(11)$$\mathrm{CD}\left(\hat{P},P\right)=\frac{1}{2n}\sum_{i=1}^{n}{∥{\hat{x}}_{i}-\mathrm{NN}\left({\hat{x}}_{i},P\right)∥}_{2}^{2}+\frac{1}{2m}\sum_{j=1}^{m}{∥x_{j}-\mathrm{NN}\left(x_{j},\hat{P}\right)∥}_{2}^{2},$$
where $\hat{P}$ and P denote the predicted and reference point clouds, and NN(·,·) returns the nearest neighbor. To evaluate the geometric consistency with the underlying surface, we also compute the P2M distance. This metric compares points to a mesh surface, taking into account both the distance from points to the mesh and vice versa. It is defined as(12)$$P2M\left(\hat{P},M\right)=\frac{1}{2n}\sum_{i=1}^{n}\min_{f\in M}d\left({\hat{x}}_{i},f\right)+\frac{1}{2|M|}\sum_{f\in M}\min_{{\hat{x}}_{i}\in \hat{P}}d\left({\hat{x}}_{i},f\right),$$
where M denotes the ground-truth mesh, and d(x,f) measures the shortest distance between a point and a mesh face. The first term captures how well the predicted points lie on the mesh surface, while the second encourages surface coverage. All point clouds and meshes are normalized to the unit sphere before evaluation to ensure scale invariance.

### 4.3. Training Details

Training is conducted on a single NVIDIA H100 GPU 80 GB with an Intel(R) Xeon(R) Platinum 8480+ CPU, running Ubuntu 22.04.2 LTS. The model is trained for a total of 650,000 iterations with a batch size of 32. Automatic mixed precision is enabled for memory and computing efficiency, and gradient clipping with a maximum norm of 1.0 is applied to stabilize training. Both the denoiser network and the discriminator of ADBM are trained using the AdamW optimizer. The denoiser network training uses a constant learning rate of 0.0003, while the discriminator is trained with a learning rate of 0.0001. The exponential moving average of the denoiser network parameters is maintained with a decay factor of 0.999. We use 10 reverse diffusion steps during both adversarial training and evaluation to generate denoised point clouds.

### 4.4. Experimental Results

We evaluate our method, ADBM, on the PU-Net and PC-Net datasets under varying Gaussian noise levels and point cloud resolutions. Table 1 presents the quantitative comparison of the denoising performance based on Chamfer distance and Point-to-Mesh distance, where lower values indicate better denoising performance. On the PU-Net dataset with 10 k input points, ADBM consistently outperforms all baselines across all noise levels. At 1% noise, ADBM records a CD of 2.18 and a P2M of 0.34, outperforming P2P-Bridge which achieves 2.45 for CD and 0.39 for P2M. When the noise level increases to 2%, ADBM achieves 3.15 for CD and 0.77 for P2M, showing improvements over P2P-Bridge’s 3.27 and 0.86, respectively. At the highest noise level of 3%, ADBM achieves 3.98 for CD and 1.40 for P2M, compared to 4.07 and 1.47 by P2P-Bridge. For the high-resolution setting with 50 k points, ADBM continues to outperform the baselines. At 1% noise, ADBM achieves a CD of 0.57 and P2M of 0.08, showing improvements over P2P-Bridge’s values of 0.60 and 0.09. At 2% noise, the CD and P2M values achieved by ADBM are 0.90 and 0.32, respectively, whereas P2P-Bridge achieves 0.95 and 0.35. At 3% noise, ADBM yields 1.61 for CD and 0.88 for P2M, outperforming P2P-Bridge’s values of 1.63 and 0.90. ADBM shows average relative improvements of 5.63% for CD and 9.35% for P2M in the 10 k point setting, and 3.83% and 7.30% in the 50 k point setting compared to P2P-Bridge.

On the PC-Net dataset, which is used to evaluate generalization to unseen shapes, our method, ADBM, also shows robust performance. At 10 k input points and 1% noise, ADBM records a CD of 2.82 and a P2M of 0.59, slightly improving upon P2P-Bridge’s results of 2.87 and 0.63. For 2% noise, ADBM achieves 4.43 for CD and 0.86 for P2M, again outperforming P2P-Bridge, which reports 4.52 and 0.92. At 3% noise, ADBM shows a clear advantage with a CD of 5.57 and a P2M of 1.27, while P2P-Bridge reports 5.65 and 1.34. For the 50 k-point resolution, the same trend holds. At 1% noise, ADBM achieves a CD of 0.90 and a P2M of 0.11, whereas P2P-Bridge reports 0.92 and 0.12. With 2% noise, ADBM records 1.37 for CD and 0.25 for P2M, improving upon P2P-Bridge’s 1.39 and 0.26. At 3% noise, ADBM achieves 2.14 for CD and 0.49 for P2M, while P2P-Bridge results in 2.17 and 0.51. The improvements are smaller but consistent, with 1.72% for CD and 6.03% for P2M for 10 k points, and 1.66% and 5.37% for 50 k points.

These comprehensive results demonstrate that our proposed method not only consistently outperforms the existing baselines across all noise levels and resolutions, but also generalizes effectively to unseen object categories, yielding the best performance in terms of both point-wise accuracy and surface-level fidelity. Previous denoising methods, such as ScoreDenoise [16], primarily rely on loss functions focused on noise prediction, which emphasize overall noise suppression rather than fine-grained geometric reconstruction. In contrast, the proposed method incorporates a discriminator-based adversarial loss, which explicitly enforces structural fidelity by distinguishing between clean and denoised point clouds. From a training robustness perspective, this adversarial term acts as a regularizer, guiding the model to preserve sharp edges and recover challenging geometric patterns. As a result, the proposed method demonstrates improved performance in scenarios with diverse noise levels and intricate structural shapes, where conventional score-based approaches may struggle.

To further investigate the generalization behavior of the proposed method, we conducted a class-wise performance analysis on both the PU-Net and PC-Net datasets. Figure 3 presents the CD and P2M metrics for each class under the 10 k point and 1% Gaussian noise setting. The results reveal that reconstruction difficulty varies substantially across object categories, with geometrically complex structures (e.g., chair, elk) exhibiting higher error values. In contrast, objects with smoother or more compact surfaces tend to yield lower reconstruction errors, reflecting the relative ease of recovering their geometric details. Notably, the model maintains competitive performance across unseen PC-Net shapes, indicating robust generalization to novel object geometries. These observations indicate that the model effectively captures transferable structural priors rather than overfitting to the training distribution.

To qualitatively evaluate the denoising performance, Figure 4 presents visual comparisons across various object categories. The first row shows the ground-truth clean point clouds, uniformly sampled with 10 k points per object. To generate the noisy inputs shown in the second row, Gaussian noise with a standard deviation of 1% unit sphere radius is added to the clean shapes. These noisy point clouds exhibit substantial structural distortion and irregular point distribution, particularly around thin or intricate regions such as the camel’s legs, the chair’s backrest, and the curvature of the duck shape. The third row shows the outputs produced by the P2P-Bridge baseline without adversarial learning. While the overall shapes are recovered to some extent, the results often suffer from blurring or loss of fine details. For instance, the legs of animal models appear less distinct, and the duck’s bill lacks geometric sharpness and continuity. In comparison, the proposed method, in the fourth row, restores both global structure and fine-grained geometric details. The denoised results exhibit more faithful alignment with the ground truth, better preserving object-specific characteristics and surface continuity. Moreover, the point distribution appears more uniform and natural, indicating improved surface coverage and sampling quality. These qualitative observations are consistent with the quantitative results, highlighting the superior denoising capability and structural fidelity of our method across diverse shapes.

Figure 5 shows per-point error heatmaps between the denoised outputs and the ground-truth shapes, where the color represents the Euclidean distance to the corresponding ground-truth point. All samples consist of 10 k points, and the input noise follows a Gaussian distribution with a standard deviation of 1% of the unit sphere. Overall, our method achieves low reconstruction errors across most surface regions, especially in smooth and planar areas such as the camel’s torso and the cow’s flank. These regions are predominantly rendered in blue, indicating accurate point-wise recovery. However, increased reconstruction errors are observed in geometrically complex areas, including thin structures and high-curvature boundaries, such as for the camel’s legs, the edges of the chair’s backrest, and the tail of the horse. These failure cases typically arise due to the local sparsity or overlapping noise in the input, which can distort fine geometric cues during denoising. To mitigate these localized failures, future work may focus on stabilizing the adversarial training process and improving the loss function to better capture fine-grained geometric discrepancies. In particular, incorporating region-aware weighting schemes or multi-scale structural constraints into the training objective could enhance the model’s sensitivity to delicate features. These improvements may lead to more faithful reconstructions in challenging regions.

### 4.5. Ablation Results

We conducted an ablation study to investigate the effect of the adversarial loss weight $\lambda_{\mathrm{adv}}$ on the denoising performance, as summarized in Table 2. Across different Gaussian noise levels and point counts, the proposed method consistently improved results compared to the base model without adversarial learning. Among the tested values, $\lambda_{\mathrm{adv}}=0.7$ yielded the best overall performance, achieving the lowest CD and P2M errors for most settings. While $\lambda_{\mathrm{adv}}=0.5$ sometimes produced competitive results, especially with 50 k points and the 3% Gaussian noise setting, its performance degraded under other scenarios. $\lambda_{\mathrm{adv}}=0.9$ showed no clear advantage and in some cases slightly worsened the results, suggesting training instability of the adversarial component. Based on the results, we selected $\lambda_{\mathrm{adv}}=0.7$ as the optimal trade-off between shape fidelity and adversarial guidance, leading to robust denoising performance across diverse noise levels and point densities.

## 5. Conclusions

In this paper, we proposed an adversarial diffusion bridge training method for 3D point cloud denoising. Building on the Schrödinger Bridge formulation, our method models the interpolation between noisy and clean point clouds, enabling effective restoration of fine-grained geometry. To further improve the perceptual quality and fidelity of denoised outputs, we introduced an adversarial learning scheme, where a lightweight discriminator is trained to guide the generator toward producing samples indistinguishable from real clean point clouds. The proposed method achieves superior reconstruction fidelity, showing strong generalization performance across diverse object categories. However, as shown in Figure 5, denoising performance in highly corrupted or geometrically complex regions remains challenging. These cases highlight the need for further refinement of the adversarial component. In future work, we aim to explore improved training stability through adversarial loss regularization and conduct systematic studies on how varying the weighting parameter (e.g., $\lambda_{\mathrm{adv}}$) influences the denoising quality and convergence behavior.

## Figures and Tables

**Figure 1 sensors-25-05261-f001:**
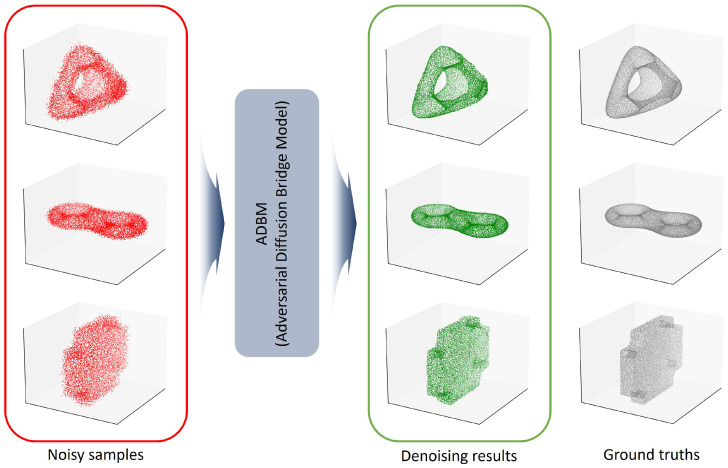
Visual examples of point cloud denoising results using the proposed method, ADBM.

**Figure 2 sensors-25-05261-f002:**
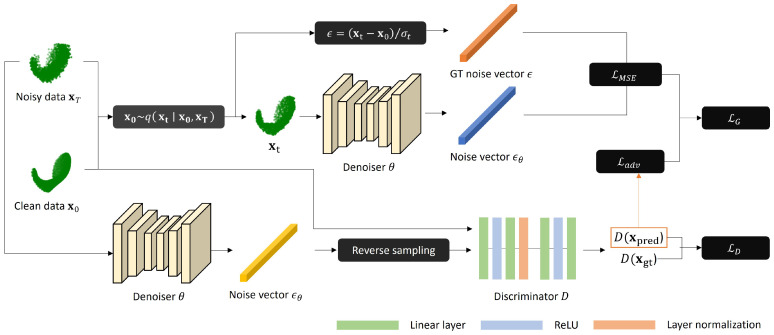
Overview of the proposed adversarial diffusion bridge model (ADBM) training pipeline.

**Figure 3 sensors-25-05261-f003:**
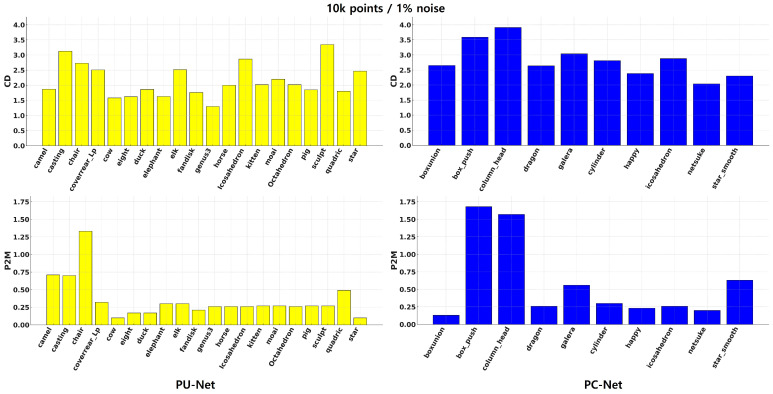
Class-wise denoising performance (CD↓ / P2M↓) on PU-Net and PC-Net datasets with 10 k points and 1% Gaussian noise.

**Figure 4 sensors-25-05261-f004:**
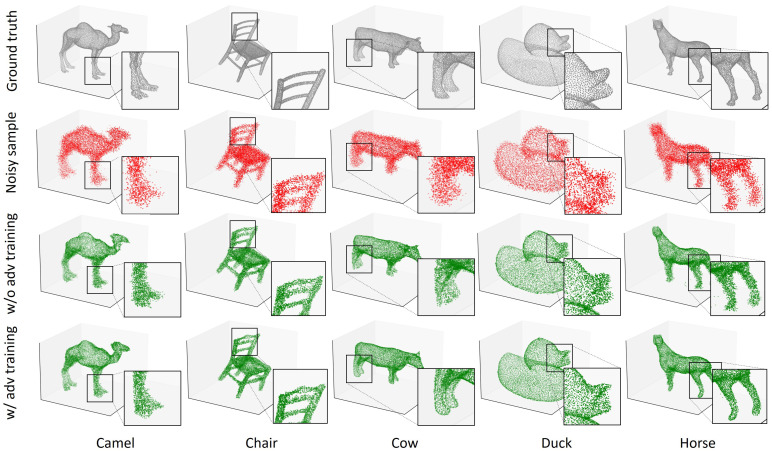
Qualitative comparison of point cloud denoising results with close-up views of key object regions (e.g., legs, backs, and bills).

**Figure 5 sensors-25-05261-f005:**
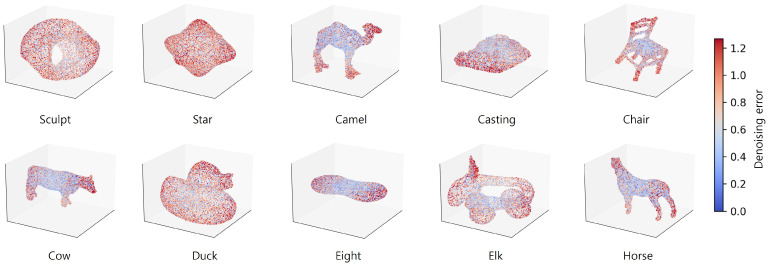
Visualization of per-point Euclidean errors between the denoised outputs and ground-truth point clouds.

**Table 1 sensors-25-05261-t001:** Comparison of denoising performance (CD↓ / P2M↓) under different Gaussian noise levels and point counts. The best results are highlighted in bold.

Dataset	Number of Points Gaussian Noise Level Method/Metric	$10\cdot 10^{3}$ Points						$50\cdot 10^{3}$ Points					
		**1%**		**2%**		**3%**		**1%**		**2%**		**3%**	
		**CD**	**P2M**	**CD**	**P2M**	**CD**	**P2M**	**CD**	**P2M**	**CD**	**P2M**	**CD**	**P2M**
PU-Net [26]	PC-Net [15]	3.52	1.15	7.47	3.97	13.1	8.74	1.05	0.35	1.45	0.61	2.29	1.29
	ScoreDenoise [16]	2.52	0.46	3.69	1.07	4.71	1.94	0.72	0.15	1.29	0.57	1.93	1.04
	P2P-Bridge [17]	2.45	0.39	3.27	0.86	4.07	1.47	0.60	0.09	0.95	0.35	1.63	0.90
	ADBM (ours)	**2.18**	**0.34**	**3.15**	**0.77**	**3.98**	**1.40**	**0.57**	**0.08**	**0.90**	**0.32**	**1.61**	**0.88**
PC-Net [15]	PC-Net [15]	3.85	1.22	6.04	1.45	5.87	1.29	0.29	0.11	0.51	0.25	3.25	1.08
	ScoreDenoise [16]	3.37	0.95	4.52	1.16	6.78	1.94	1.07	0.17	1.66	0.35	2.49	0.66
	P2P-Bridge [17]	2.87	0.63	4.52	0.92	5.65	1.34	0.92	0.12	1.39	0.26	2.17	0.51
	ADBM (ours)	**2.82**	**0.59**	**4.43**	**0.86**	**5.57**	**1.27**	**0.90**	**0.11**	**1.37**	**0.25**	**2.14**	**0.49**

**Table 2 sensors-25-05261-t002:** Ablation study on the adversarial loss weight $\lambda_{\mathrm{adv}}$ for the PU-Net dataset under varying Gaussian noise levels and numbers of points. The best results are highlighted in bold.

Number of Points	$10\cdot 10^{3}$ Points						$50\cdot 10^{3}$ Points					
**Gaussian Noise Level**	**1%**		**2%**		**3%**		**1%**		**2%**		**3%**	
$\lambda_{\mathrm{adv}}$ **/Metric**	**CD**	**P2M**	**CD**	**P2M**	**CD**	**P2M**	**CD**	**P2M**	**CD**	**P2M**	**CD**	**P2M**
Base (w/o ADBM)	2.45	0.39	3.27	0.86	4.07	1.47	0.60	0.09	0.95	0.35	1.63	0.90
0.5	2.28	0.38	3.28	0.85	4.06	1.46	0.59	0.09	0.91	0.34	**1.54**	**0.82**
0.7	**2.18**	**0.34**	**3.15**	**0.77**	**3.98**	**1.40**	**0.57**	**0.08**	**0.90**	**0.32**	1.61	0.88
0.9	2.30	0.38	3.32	0.87	4.10	1.47	0.60	0.09	0.97	0.37	1.70	0.95

## Data Availability

Not applicable.

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
