# Peer review of "ADBM: Adversarial Diffusion Bridge Model for Denoising of 3D Point Cloud Data"

_sensors, 2025, doi:10.3390/s25175261_

Round 1
Reviewer 1 Report
Comments and Suggestions for Authors
The paper ``ADBM: Adversarial Diffusion Bridge Model for Denoising of 3D Point Cloud Data'' proposes a combination of diffusion bridge models with adversarial training for 3D point cloud denoising. The authors have trained and tested this method on the PU-Net and PC-Net datasets. While the results appear promising, several issues need to be addressed:
1. It is recommended that the authors clearly enumerate the innovative contributions of this work relative to the existing literature in the introduction section.
2. The point cloud renderings in Figure 3 are too small to subjectively evaluate the denoising effects. Please provide 300 dpi close-up images of specific areas (such as chair backs, camel legs, camel humps, cow horns, duck bills, etc.) to facilitate accurate assessment of the authors' work.
3. The authors emphasize the lightweight design of the discriminator and mention computational memory and efficiency in Section 4.3, but no specific quantitative values are provided in the paper.
4. There is a notable lack of ablation studies, and only a single result with λadv = 0.7 is presented, which is insufficient to demonstrate the rationality of this hyperparameter setting. The discussion should be expanded to address how λadv affects potential gradient instability or detail distortion phenomena.

Reviewer 2 Report
Comments and Suggestions for Authors
The paper proposes an Adversarial Diffusion Bridge Model. The idea is original and builds convincingly on the limitations of existing diffusion models by enhancing geometric detail recovery under noise. However, some aspects could be improved before its publication:
- Lack of Statistical Significance Analysis. There is no mention of standard deviation, confidence intervals, or statistical tests. Include error bars or perform a statistical significance test to demonstrate whether the improvements over baselines are significant and consistent across multiple runs.
- Limited Dataset Generalization Analysis. PC-Net is only used for testing, but the paper lacks deeper analysis on why the model generalizes well or where it fails. Include class-wise results or t-SNE visualizations to show embedding separability and better explain generalization behavior.
- Missing Ablation Studies. No ablation is presented to show the impact of the adversarial component (λ_adv) vs. base model (pure diffusion)
- Provide more discussion on why ADBM outperforms ScoreDenoise in specific scenarios (is it due to better geometry reconstruction, or more robust training?
- Include comparisons of model complexity, FLOPs, or FPS during inference to highlight practicality.
- Explain acceptable thresholds or provide percentage improvement over baseline.
Round 2
Reviewer 2 Report
Comments and Suggestions for Authors
The article has improved after the reviews, I find it suitable for publication